# New Biophysical Approaches Reveal the Dynamics and Mechanics of Type I Viral Fusion Machinery and Their Interplay with Membranes

**DOI:** 10.3390/v12040413

**Published:** 2020-04-08

**Authors:** Mark A. Benhaim, Kelly K. Lee

**Affiliations:** 1Department of Medicinal Chemistry, University of Washington, Seattle, WA 98195-7610, USA; mbenhaim@uw.edu; 2Biological Physics Structure and Design Program, University of Washington, Seattle, WA 98195-7610, USA

**Keywords:** membrane fusion, viral fusion, type I fusion protein, influenza, hemagglutinin, dynamics, structural mechanics, biophysics, mechanisms of membrane fusion

## Abstract

Protein-mediated membrane fusion is a highly regulated biological process essential for cellular and organismal functions and infection by enveloped viruses. During viral entry the membrane fusion reaction is catalyzed by specialized protein machinery on the viral surface. These viral fusion proteins undergo a series of dramatic structural changes during membrane fusion where they engage, remodel, and ultimately fuse with the host membrane. The structural and dynamic nature of these conformational changes and their impact on the membranes have long-eluded characterization. Recent advances in structural and biophysical methodologies have enabled researchers to directly observe viral fusion proteins as they carry out their functions during membrane fusion. Here we review the structure and function of type I viral fusion proteins and mechanisms of protein-mediated membrane fusion. We highlight how recent technological advances and new biophysical approaches are providing unprecedented new insight into the membrane fusion reaction.

## 1. Introduction

The process of protein-mediated membrane fusion is essential for a range of cellular and organismal functions. It is involved in synaptic signaling, cellular communication, intra- and extra-cellular vesicle trafficking, mitochondrial homeostasis, sexual reproduction, embryogenesis, and infection by enveloped viruses [1,2,3,4,5,6]. Infection by all enveloped viruses requires fusion of the viral and host membranes in order to deliver the viral genome and replication machinery across the host cell membrane to a suitable subcellular location and initiate an infection cycle. Enveloped viruses have evolved specialized protein machinery that drive this process to completion by undergoing a series of large-scale conformational changes [7]. The nature of these changes and the resulting impact on the membranes themselves have long-eluded characterization, but new biophysical techniques are providing a detailed glimpse into dynamic changes in the protein machinery as well as revealing new structural and mechanistic insights into the interplay of proteins and membranes during fusion.

All viral fusion proteins encode the same basic functionality: activate in response to a specific trigger(s), engage the target host membrane, draw the host membrane into close apposition with the viral membrane, and induce the membranes to merge. Viral fusion proteins are classified into three distinct classes, with the type I fusion proteins being the best characterized to date [1,4,7,8]. Among the type I proteins, the influenza virus hemagglutinin (HA) is the most widely studied and has served as the foundation upon which much of our understanding of viral membrane fusion proteins has been built. Type II fusion proteins are found in viruses including flaviviruses and alphaviruses such as Dengue, Zika, and Chikungunya viruses, and even have been identified in eukaryotic cell-cell fusion systems [9,10,11,12]. Type III fusion proteins are found in rhabdoviruses (such as Rabies and Vesicular Stomatatis virus G glycoproteins), herpesviruses (Herpes Simplex virus 1 gB protein), as well as baculovirus [12,13,14,15,16]. While the individual folds exhibited by these classes are completely different, they share common functional traits in that all adopt a pre-fusion conformation prior to activation in which one terminus of the protein is anchored in the virus membrane by a transmembrane domain and a second membrane active component, either a fusion peptide or loop, is sequestered from interacting with membranes (Figure 1) [12]. A trigger or set of triggers, such as exposure to low pH in endosomes or receptor binding, spurs the machinery to reorganize into a post-fusion state in which the two membrane active components are colocalized.

Until recently, static structures of the pre- and post-fusion states of isolated fusion protein ectodomains and biochemical or spectroscopic measurements were the primary pieces of information that informed our models of membrane fusion. While the static structures provide defined endpoints for the conformational change that drives membrane fusion, they do not tell us how these conformational changes occur or how these proteins interact with and perturb the lipid membrane during fusion. Likewise, fluorescence spectroscopy and circular dichroism studies have shown that HA-fusion activation leads to population of discernable intermediates rather than transitioning directly and irreversibly from pre- to post-fusion states [17,18,19,20]. These studies show that not all HAs respond to activation in the same way and some require different pH conditions to fully activate [20]. While such studies provided valuable information on how influenza HA responds to activation conditions, they could not resolve the structure of intermediates or the specific conformational changes that occur.

A defining characteristic of type I and II viral fusion proteins is that the pre-fusion conformation is trapped in a high-energy, metastable state with respect to the low-energy, post-fusion conformation [4,7,21]. Once triggered, this energetic imbalance ultimately results in the fusion protein undergoing an irreversible transition to the post-fusion state. Indeed many early pioneering studies on influenza HA led to the development of the “spring-loaded” mechanistic model for viral membrane fusion that is the prevailing way in which these machines are considered to function [21,22,23,24,25,26,27,28,29,30,31,32]. In this model, type I fusion proteins function analogous to taut springs that are poised in a high energy state that, once triggered, rapidly and irreversibly “spring” or refold to the low-energy, post-fusion state. The type III rhabdovirus G proteins are an exception to this general trend and exhibit reversible pH-dependent conformational switching [13,15].

While membrane fusion is a thermodynamically favorable process, it requires an input of free energy to dehydrate the phospholipid headgroups as they are drawn into close apposition during fusion [2,21,33,34]. This repulsive “hydration force” presents a kinetic barrier to fusion that prevents spontaneous, aberrant fusion events from occurring. This renders the membrane fusion reaction a tightly controlled biological process [12,35,36,37]. The fusion peptides (type I) or fusion loops (type II) are believed to facilitate this reorganization of bound water while the free energy released from the exothermic refolding of the fusion proteins from the metastable pre-fusion to the low-energy post-fusion state is harnessed to remodel, induce curvature and local defects in the lipid bilayer, and drive the membranes together [12,21,37,38,39,40,41,42].

Direct structural characterization of viral membrane fusion and the machinery involved has been impeded by the dynamic nature of the membrane fusion reaction. To understand the process of protein-mediated membrane fusion, it is necessary to characterize the intermediate states that are traversed both at the level of the protein machinery and the organization of the lipid membrane itself. Recent technological advances, notably those involving cryo-electron microscopy (cryo-EM), have enabled the structure of many viral fusion proteins to be imaged at high resolution [4,43,44,45]. These structures revealed that remarkably similar features were present in the fusion proteins of diverse viruses [4,46]. Even more exciting in terms of finally being able to dissect mechanisms of this complex biological process is that recent developments in cryo-electron tomography (cryo-ET), single molecule FRET (sm-FRET), and structural mass spectrometry have enabled the direct monitoring of viral membrane fusion and dissection of the conformational changes that take place in the fusion machinery during fusion [43,44,45,46,47,48,49,50,51,52,53,54,55,56]. Here, we review our current understanding of the structure and function of type I viral fusion proteins, the structural mechanics of fusion protein activation, and the mechanism of protein-mediated viral membrane fusion.

## 2. Structural Organization of Class 1 Viral Fusion Proteins

Type I viral fusion proteins are homotrimeric glycoproteins that decorate the viral envelope. These proteins are synthesized as inactive, single-chain, polypeptide precursors that assemble into trimers and are proteolytically processed by host cell proteases into their functional, metastable pre-fusion states [4,7,57,58,59]. Proteolytic processing can take place during viral assembly, maturation, and/or entry depending on the specific virus. In the case of influenza, the uncleaved, fusion-incompetent glycoprotein trimer, HA0, is primarily processed by extracellular trypsin-like proteases after new virions are released from infected cells [23,26,32,57,58,60]. HA’s from highly pathogenic avian influenza viruses often contain a polybasic motif at their cleavage site and are processed by endogenous proteases, such as furin, in the trans-Golgi network [61,62,63]. The resulting functional HA assembly is a homotrimer of disulfide-linked heterodimers consisting of a receptor binding subunit, HA1, and a membrane fusion subunit, HA2 (Figure 1A). The newly formed N-terminus of HA2 includes the highly conserved hydrophobic fusion peptide which becomes sequestered in a pocket within the central helical bundle of HA2 in the pre-fusion conformation (Figure 1A) [24,26,30,32,64]. The C-terminus of HA2 is anchored in the viral membrane by a helical transmembrane domain [65]. The HA1 globular head contains a sialic acid receptor binding site positioned at the apex of the trimer in the pre-fusion conformation [24].

Infection by influenza virus begins when HA binds sialic acid on cell surface receptors through a low-affinity, high-avidity interaction, triggering uptake into cells by receptor-mediated endocytosis or macropinocytosis [32]. As the endosomal lumen becomes increasingly acidic, low pH triggers a cascade of conformational changes throughout the HA that culminate in the irreversible reorganization to the post-fusion conformation and fusion of the viral and endosomal membranes (Figure 1A) [25].

The conventional mechanistic model describing HA’s membrane fusion activity suggests that two dominant stabilizing interactions within HA, termed the “clamp” and “hook”, act to maintain the metastable pre-fusion conformation [21]. In this model, the HA1 globular head acts as a stabilizing “clamp” on the high-energy and spring-loaded HA2 fusion domain. The HA1 globular head rests atop the HA2 apex where HA1 forms stabilizing contacts with the HA2 B-loop (Figure 1A). N- and C-terminal segments of HA1 form extended quaternary contacts with the HA2 A-helix and B-loop and are likely important for maintaining the pre-fusion conformation (Figure 1A) [21,31]. The sequestered N-terminal HA2 fusion peptide forms a “hook”, lashing the central helices of adjacent HA2 protomers together. According to this model, once activated by low pH, these interactions become destabilized and release the high-energy spring-loaded HA2 fusion domain which rapidly and irreversibly reorganizes to the post-fusion state.

Comparison of the pre- and post-fusion crystal structures of the HA ectodomain reveals the dramatic conformational changes that occur as a result of this reorganization (Figure 1A). In the post-fusion state, the central B-loop segment of HA2 has converted from an extended coil to a helix that extends the central helical bundle [31]. As a result, the fusion peptide is projected towards the target host membrane. Structures of post-fusion HA2 have revealed that C-terminal portions of the subunit also refold. In this case, a helix converts to a turn, which enables the C-terminal “leash” attached to the viral membrane anchor to run along the groove formed by the helical bundle. This “leash-in-the-groove” interaction has been shown to be necessary for drawing the two membrane active components together, leading the viral and target membranes into close contact and inducing them to merge [66]. These structures provided the beginning and endpoints of the pathway the fusion machinery takes, but neither in fact correspond to the fusion-active forms of the trimer that manipulate the membranes, nor do they reveal the pathway of conformational change that links the end states.

## 3. A Conserved Architecture Shared by Divergent Viruses

Pre- and post-fusion structures of many type I viral fusion proteins from diverse viruses have been solved (Figure 1A,B) [1,4,7]. To date, pre- and post-fusion structures have been determined for the influenza HA, human immunodeficiency virus (HIV-1) Env, Coronavirus (CoV) S, Ebola virus GP, Lassa virus GPC, Parainfluenza virus (PIV5) F, and respiratory syncytial virus (RSV) F proteins (Figure 1A,B) [67,68,69,70,71,72,73,74,75,76]. While the pre-fusion structures of these diverse fusion proteins differ in size and elaborations, they share a conserved organization and architecture of their fusion subunits which feature two central heptad repeats that reorganize to form the three hairpins at the core of the six-helix bundle in the post-fusion state, bringing the viral and host target membranes together (Figure 1A,B) [7,12]. Furthermore, in the pre-fusion state the fusion subunits for these viral fusion proteins exhibit extensive interactions with the head domains that, as for influenza HA, could be interpreted to help “clamp” the fusion subunit in its pre-triggered conformation (Figure 1A,B).

## 4. Despite their Common Architectures, Activation Mechanisms and Triggers are Highly Divergent Among Type I Fusion Proteins

The process of fusion protein activation and the means by which the fusion trigger is communicated across domains are not well understood for the majority of type I viral fusion proteins. Indeed, for the best-characterized system, HA, structural and biophysical data that reveal details of fusion activation and membrane fusion have only recently become available.

To date, technical limitations have hindered researchers’ abilities to directly observe fusion intermediates with adequate resolution. Indeed, even in the relatively well-characterized HA case, a range of rather different models of fusion protein activation and conformational change have been proposed. One of the earliest and most informative studies performed by White and Wilson, used a panel of antibodies against HA to determine the sequence of early structural rearrangements that occur during HA-fusion activation [27]. This approach, however, was limited to resolving changes significant enough to expose the antibody epitope. Indeed, antibody probes have the potential to perturb the behavior of the system due to their strong interactions with the antigen and large size. Despite these limitations, White and Wilson concluded that HA exhibits two sequential conformational changes during fusion activation: reorganization of the HA2 stem region followed by a dissociation of the HA1 globular head domains leading to opening of the HA apex. Furthermore, White and Wilson demonstrated that activation of the soluble bromelain-released HA ectodomain (BHA) was slightly faster than detergent-solubilized full-length HA, suggesting that the intact trimer behaves differently than the ectodomain alone. This study was among the first to suggest that HA undergoes a sequence of conformational changes, transiently populating intermediate states, along the fusion pathway.

The antibody-monitored changes in the HA structure seemed to contradict an “HA1 uncaging” model for hemagglutinin activation that suggests this fusion protein’s activation is initiated by the dissociation of the HA1 globular head domains, which would necessarily precede HA2 triggering. The uncaging model is supported by data showing that dissociation of the HA1 globular head is essential for HA’s membrane fusion activity [27,28,29,77]. Expression of HA2 in the absence of HA1 yields HA2 in its post-fusion conformation [27,28,29,77]. Furthermore, preventing HA1′s dissociation either through the introduction of interprotomer disulfide bonds or antibody binding renders HA non-fusogenic, but these restraints still enable the HA2 fusion peptide to release and interact with target membranes [28,29,77,78,79].

Ultimately, to understand HA’s native modes of activation, it is necessary to directly probe the sequence of conformational changes that occur during HA-fusion activation. Recent direct observations of HA fusion intermediates and the membrane fusion process, enabled by advances in cryo-EM, sm-FRET, and structural mass spectrometry have begun to challenge these long-standing conceptions about HA. In 2012 using cryo-electron tomography, Fontana et al. observed low pH-induced morphological changes in full-length HA on the virus surface that suggested reorganization of the HA2 fusion domain preceded dissociation of the HA1 globular head and that these changes were reversible, within a certain window in time, upon return to neutral pH [80]. From the tomograms, sub-tomogram averaged structures of fusion-active HA revealed morphological changes that were consistent with those suggested to occur during the so called “fusion peptide release” mechanism of HA-fusion activation, similar to those put forth by White and Wilson nearly 30 years earlier [27,78,80]. In contrast to the long-held and conventional “HA1 uncaging” model, this alternative model posits that first the HA2 fusion domain becomes activated by low pH, releasing the fusion peptide from sequestration where it is free to engage the target membrane prior to complete dissociation of the HA1 globular head [78]. While the averaged images gave a tantalizing glimpse of potential HA intermediates, they did not offer sufficient structural resolution to elucidate detailed structural changes in HA.

More recently, Garcia et al. sought to understand the dynamic structural changes that occur throughout the soluble HA ectodomain at low pH at the threshold of fusion activation using hydrogen/deuterium-exchange mass spectrometry (HDX-MS) [81]. HDX-MS is a solution state biophysical and structural technique that monitors the accessibility of amide hydrogens along the protein backbone. HDX-MS directly monitors dynamic structural changes and motion throughout a protein that are otherwise invisible to other structural approaches [45,50,82,83]. At low pH conditions approaching fusion activation, dynamic changes across the HA were observed where the HA1-HA1 trimeric interface became bolstered and the HA2 fusion peptide proximal subdomain became more dynamic. The authors concluded that at increasingly acidic condition, prior to activation, HA becomes primed for fusion peptide release, adding further support to this emerging mechanistic model. These recent studies suggest that in the early stages of fusion activation HA adopts a dynamic fusion-peptide-released intermediate state [22,27,33,80,81,84,85,86,87].

HA has served as the system against which other type I fusion proteins have been compared. Low pH activation of HA, however, is relatively simple when compared to other fusion proteins with more varied and complex activation modes [8]. Due to the complexities of the triggering mechanisms for other type I systems, comparatively less is known about how they function. Perhaps the next best characterized system after HA is HIV-1 Envelope (Env) fusion glycoprotein, which is activated by two successive receptor binding events [4,75,88,89,90,91,92]. Env first binds the CD4 receptor on the surface of T-cells which induces reorganization of the gp120 receptor binding domain and exposure of the co-receptor binding site enabling binding of either CCR5 or CXCR4 [7,92]. Recently cryo-EM, sm-FRET, and HDX-MS have been used to characterize the structure of Env in the apo and CD4-bound conformations, illuminating how CD4 binding induces long-range conformational changes throughout Env, priming it for coreceptor binding [75,88,89,90,91]. Sm-FRET revealed that even receptor-naive Env dynamically samples multiple conformations at equilibrium, including a state that seems to mimic the CD4-bound state [75,89,90,91]. While these studies have revealed valuable information about Env structural dynamics, there remains little understood about how coreceptor binding activates Env during membrane fusion.

The Lassa virus fusion glycoprotein complex (GPC), like HA, is triggered by low pH in the endosome [93,94]. However, GPC also must initially bind to two successive receptors in order for the Lassa virus to become internalized into host cells and deliver the viral replication machinery to the correct cellular compartment [93,94,95]. GPC first binds α-dystroglycan receptors on the cell surface and is internalized into endosomes [93,94,95,96]. As the endosome approaches pH 6.0, GPC dissociates from α-dystroglycan and binds the endosomal receptor LAMP-1 [94,97,98,99]. GPC is only capable of binding to LAMP-1 under acidic conditions in the endosome, and while LAMP-1 binding is not required for fusion to occur, it raises the pH of activation for GPC from pH 4.0 to above 5, increases fusion efficiency, and infectivity [94,97,98,99,100]. Cryo-ET analysis of GPC in complex with LAMP-1 at low pH suggests that the GP1 receptor binding domain (RBD) subunit reorganizes at low pH exposing the LAMP-1 binding site [99]. However, this study was unable to resolve LAMP-1-induced structural changes in GPC and thus the molecular mechanism for how LAMP-1 binding primes GPC for fusion at elevated pH conditions remains elusive [98,99,100].

In other type I fusion systems, such as those found in paramyxoviruses, receptor binding and the fusion machinery are distributed between two proteins. Each apparently undergoes conformational changes that change their mode of interaction, leading the fusion protein to become fusogenic [67,101,102]. Two models have been proposed that describe how paramyxovirus F protein activation occurs through the interaction with the receptor binding protein (HN, H, or G protein). The “dissociation” or “clamp” hypothesis suggests that the fusion and attachment proteins are associated on the viral surface before receptor binding and that this interaction acts to stabilize the F protein in the metastable pre-fusion conformation, similar to how the RBDs of other type I fusion proteins act as a “clamp” and the fusion domain [101,102]. Upon receptor binding, the attachment protein releases the fusion protein which then becomes fusion-active. Alternatively, the “provocateur” hypothesis suggests that the fusion and attachment proteins exist freely on the viral surface and, upon receptor binding, they associate, leading the F protein to become fusion-active. One key difference between these two hypotheses is that in the dissociation/clamp hypothesis the interaction between the attachment and fusion protein is stabilizing, whereas in the provocateur hypothesis this interaction acts to destabilize the F protein [101,102].

The biophysical techniques piloted in studies of HA-mediated membrane fusion may begin to reveal the nature of the conformational changes in these systems, however at present much remains to be understood about how the complex set of environmental, receptor-binding, and proteolytic processing triggers are communicated to their fusion subunit machinery.

## 5. Direct Monitoring of the Transitions between Conformational States

While atomic resolution structural models provide the highest level of detail for understanding a protein architecture, they are less suited for tracking protein dynamics and conformational change. Researchers have recently turned towards approaches such as single molecule-FRET (sm-FRET), that enable the study of proteins’ motions. Using sm-FRET, Das et al. directly observed, for the first time, an obligate and highly dynamic fusion intermediate for influenza HA [22]. By producing virus-like particles (VLPs) where, on each VLP, a single HA trimer bore one pair of FRET labels, the authors were able to monitor, in real time, the dynamic structural changes that occurred in HA2 during low-pH-induced fusion activation and membrane fusion. Their data showed that, even at neutral pH, HA was remarkably dynamic and reversibly transitioned between at least two distinct states. During fusion activation, in the absence of a target membrane, HA was observed to reversibly transition through a long-lived, obligate-fusion intermediate before irreversibly transitioning to the post-fusion conformation (Figure 2A). When a target membrane was present, HA transitioned to the same intermediate state, however, the subsequent transition to the irreversible post-fusion state was significantly faster. While this approach cannot definitively resolve the detailed structure of each state, sm-FRET monitoring yields information describing the dynamic behavior and lifetimes of each state reported by the relative positioning of the FRET dye pairs. Furthermore, by inferring the position of each FRET label based upon its attachment residue, the authors were able to develop structural models that correspond to each FRET state. The resulting mechanistic model put forth by the authors describing HA-fusion activation and membrane fusion provided a detailed glimpse into the intermediate states and transitions for this long-studied fusion protein (Figure 2B). Despite the advances, much about the molecular mechanisms and structural nature of HA-fusion activation remains poorly understood.

Previous studies on HA-fusion activation and membrane fusion had been able to resolve the structural changes that occur throughout the trimer during fusion activation. In the sm-FRET case, changes involving HA1 could not be followed since the labeling approach was limited exclusively to monitoring the relative position of the positions in HA2 labeled by the FRET pair [22]. Recently, using structural mass spectrometry, Benhaim et al. monitored the full sequence of structural changes that occur throughout HA during fusion activation using whole influenza virions [56]. A pulse-labeling HDX-MS approach enabled snapshots of the HA’s structure to be captured during fusion activation. Pulse-labeling HDX-MS monitors changes in the accessibility of amide hydrogens along the protein backbone during a protein’s conformational change. Changes in the local structure of a protein, including changes in secondary structure or quaternary organization, in most cases yield resolvable HDX states. Monitoring these changes over time elucidated the sequence of detailed conformational changes that occur in HA1 and HA2 during fusion activation [56]. The first changes observed involved concurrent reorganization of the HA1 trimeric interface and HA2 fusion peptide proximal subdomain resulting in formation of a highly dynamic fusion peptide-released ensemble of intermediate configurations (Figure 2B,C). In this intermediate ensemble, the HA2 fusion peptide proximal subdomain features highly dynamic A-helix and B-loop segments (Figure 2B,C). The HA2 B-loop’s loop-to-helix transition had long been viewed as the driving force behind formation of the HA2 extended helical intermediate and that, once freed from the HA1 “clamp”, it was believed that the B-loop would rapidly and irreversibly adopt a helical conformation and add to the core HA2 helical bundle [21,30,103]. The HDX-MS data however, for the dynamic intermediate ensemble, revealed that the A-helix and B-loop sampled diverse secondary structure from unstructured loops to highly-protected helices [56]. Consistent with this, computational studies also suggest that both the HA2 A-helix and B-loop segments can sample diverse structural states as they transition to the extended helical intermediate. Moreover, the computational modeling suggested that the transition to the extended helical intermediate is largely driven by trimerization of the A-helix and not the B-loop’s loop-to-helix transition [103].

Interestingly, in the soluble BHA ectodomain that lacks the transmembrane anchor, a direct, two-state transition from the pre- to post-fusion state was observed by HDX-MS. This striking result underscored the fact that in order to understand the mechanism of viral protein-mediated membrane fusion, it is necessary to examine how the conformational changes are carried out in the context of the complete viral system. While the HDX-MS study did not include a target membrane, the results were in agreement with the sm-FRET findings reported by Das et al. that did examine the effect of having a target membrane present [22]. Together, these studies provide insight into the mechanism of influenza-HA-fusion activation and membrane fusion (Figure 2).

As noted above, in contrast to influenza HA, other type I viral fusion systems present a more complex set of activation factors. Das et al. and Durham et al. recently sought to characterize the intrinsic structural dynamics and dynamic conformational changes that occur in another type I fusion protein, the Ebola virus GP fusion glycoprotein, resulting from a set of activating factors [104,105]. Once internalized into cells by macropinocytosis, the low pH conditions of the endosome activate cellular proteases that cleave and remove the heavily glycosylated mucin-like domain and glycan cap from GP [104,106,107,108,109]. Once cleaved, GP binds the Niemann-Pick C1 (NPC1) receptor, becomes activated, and mediates membrane fusion through a currently unknown mechanism [106,110,111].

Atomic resolution structures of the pre-fusion GP ectodomain without the mucin-like and transmembrane domains (GPΔTM) depict GPΔTM in a single, static conformation [74,106]. However, by sm-FRET Durham et al. observed GPΔTM to be highly dynamic and capable of reversibly interconverting between three distinct states, with the dominant high-FRET state corresponding to the pre-fusion state depicted in the crystal structure [105]. Similar dynamics were observed for GP without the mucin-like domain (GPΔmuc) when presented on the surface of pseudovirus particles (Figure 3A). Removal of the glycan cap (GP_CL_) resulted in a dramatic change in the equilibrium distribution of the three FRET states as well as reduced dynamics (Figure 3B). The authors concluded that removal of the glycan cap destabilizes the pre-fusion-like high-FRET state preferred by GPΔTM and GPΔmuc biasing GP_CL_ towards the intermediate-FRET state. NPC1 binding to GP_CL_ potentiated these changes, further biasing GP towards the intermediate-FRET state and lowering the frequency of dynamic transitions between states (Figure 3C). In this study, the donor and acceptor fluorophores were positioned on GP1 and GP2 so that the movement of GP1 with respect to GP2 could be monitored. The authors suggest that removal of the glycan cap results in a repositioning of GP1 that favors NPC1 binding and relieves conformational restrictions on GP2, conferring increased flexibility and mobility to GP2 and the fusion loop [104,105]. Thus, the reversible structural changes and dynamics observed here likely correspond to a repositioning of GP1 with respect to GP2 where GP1 does not dissociate from GP2 upon receptor binding. It is important to note that while removal of the glycan cap and NPC1 binding are necessary, they alone are not sufficient to trigger GP-mediated membrane fusion [104,105].

Additional factors implicated in Ebola entry and GP-mediated fusion include endosomal pH and exposure to Ca^2+^ [112]. In a second study of Ebola GP using sm-FRET, Das et al. monitored dynamic structural changes in the GP2 fusion domain that were induced by NPC1 receptor binding, exposure to low pH, and Ca^2+^ [104]. The authors observed that receptor binding, low pH, and Ca^2+^ act synergistically to promote membrane fusion. Furthermore, low pH and Ca^2+^ induced dynamic and reversible conformational changes in GP that prime GP for NPC1 binding. Once bound to the receptor, GP2 transitioned irreversibly to a conformation that was consistent with the post-fusion state. Thus, the authors conclude that low pH and endosomal Ca^2+^ act to prime GP for receptor binding following removal of the glycan cap by promoting transition to the receptor-binding-competent intermediate state while maintaining reversibility. Similarly, as observed by Durham et al., removal of the glycan cap resulted in a repositioning of GP1 with respect to GP2 and increased mobility in the GP2 fusion loop [104,105]. Together these results indicate that removal of the glycan cap, low pH, and Ca^2+^ act synergistically to promote an intermediate state primed for receptor binding and fusion activation [104,105]. Furthermore, this suggests the glycan cap functions not just to conceal the NPC1 binding site, but also plays a critical role in regulating the pre-fusion conformational dynamics of GP. At low pH with Ca^2+^, NPC1 binding induces reorganization of GP2 into a fusion-active intermediate state and the subsequent transition to the irreversible post-fusion state.

These studies presented the first direct evidence for how multiple fusion activation factors serve to regulate and prime a fusion protein’s activity [104,105]. While the detailed molecular mechanisms underlying these observed conformational changes remain to be understood, the study of Ebola GP further highlighted the power and versatility of the sm-FRET approach and demonstrated how this approach can be used to better understand the dynamic mechanisms of fusion-protein activation beyond the simpler systems such as influenza HA. This approach thus appears to be well-suited for use in studies of fusion proteins with complex and multicomponent activation mechanisms such as SARS, CoV-S, and Lassa GPC [59,69,93,94,95,96,97,98,99,100].

## 6. Visualizing Viral Membrane Fusion in Action

Thus far we have reviewed the structure of type I viral fusion proteins and the conformational changes that occur during the membrane fusion reaction. However, these topics have largely been discussed outside the context of the actual membrane fusion reaction. While the mechanics of viral fusion proteins, namely influenza HA, have been intensively studied, the biophysical and structural mechanics of the membranes themselves have eluded characterization. The role of the viral fusion protein in the membrane fusion reaction can be distilled, quite simply, down to: engaging the host target membrane, generating the required free energy through structural reorganization to be able to bring the two membranes into close apposition, perturb the membranes, and induce them to merge. The conventional model for influenza virus HA-mediated membrane fusion suggests that HA deforms the membranes while bringing them into close contact, resulting in formation of the hemifusion state, where the outer leaflets of each membrane have joined, and the inner leaflets remain separate [7,37,48,113,114,115,116,117,118,119,120,121]. How HA mediates formation of the hemifusion state and how hemifusion proceeds to a fusion pore is not well understood, however. Furthermore, until recently it was not known which membrane (that of the virus, or cell, or both) was being primarily perturbed and remodeled during fusion [113,122,123,124]. Direct structural characterization of the membrane fusion reaction and elucidation of the sequence of membrane remodeling by the fusion proteins has only recently become possible [43,48,49,53,113,125]. Cryo-ET in particular is uniquely suited for the direct imaging of protein and membrane structural changes during protein-mediated membrane fusion [47,48,53,55,80,113,114,115,120,125,126].

The power and utility of this approach was demonstrated by in 2010 where the ultrastructure of influenza virus membrane fusion intermediates was imaged using whole virions and synthetic membrane vesicles [113]. The cryo-ET images suggested that fusion initiates when fusion-active HA, after grappling to the target membrane and upon refolding to a post-fusion hairpin configuration, creates highly curved, localized dimples in the target membrane as it is drawn towards the more rigid, matrix-protein-reinforced, viral membrane. Density surrounding the dimples corresponded to a set of 2-8 HAs that coordinated the junction between membranes. This figure was in good agreement with previous findings that estimated the stoichiometry of viral fusion proteins required for membrane fusion [42,127,128,129,130,131,132]. The cryo-tomograms conclusively showed that during membrane fusion the viral membrane remains largely unperturbed, under the mildly acidic pH conditions examined, due to the influenza M1 presence of an intact matrix protein layer [113]. Thus, the majority of membrane remodeling at that early stage is focused on the target membrane. Only when acidic pH was further lowered, did the M1 layer dissociate from the viral membrane as would need to occur to free the lipid bilayer to complete fusion during the late stages of fusion. This study was among the first to directly image the membrane ultrastructure during fusion.

The advent of the direct electron detector for use in cryo-electron microscopy, which afforded greater sensitivity and the ability to correct for sample blurring due to beam-induced sample movement and mechanical drift, enabled high-resolution information to be retained when imaging biological complexes while limiting sample degradation from high electron exposures. For studies of membrane fusion, it became possible, for example, to consistently resolve the individual membrane leaflets of a lipid bilayer [115]. Using cryo-ET, Gui et al. sought to sequence the influenza virus membrane fusion reaction and characterize the membrane ultrastructure at each stage of the fusion process using multiple pH conditions and varied target membrane compositions [115]. The authors identified and characterized the interactions between influenza virions and liposomes and monitored the population of fusion intermediate states over time (Figure 4).

In addition to the initial point-like contact mediated by a small number of HA trimers, intermediates formed by extended regions of membranes in direct contact with each other emerged prior to formation of fusion pores that allowed transfer of the viral RNP segments into the merged virus-liposome vesicles (Figure 4). Through analysis of the population kinetics, the putative sequence of intermediate states traversed during fusion was inferred. Interestingly, in that study, hemifusion was very rarely observed—it was concluded that the hemifusion state is likely unstable and transiently populated during fusion [115]. These results challenge prior observations of the hemifusion state in past studies, which relied primarily on fluorescence-monitored membrane fusion with cell surface-expressed HA, which may not replicate the density and organization of HA on virions and also lack the important M1 matrix layer [116,117,118]. Some studies have suggested that hemifusion represents an unproductive off-pathway state for protein-mediated membrane fusion [133,134]. While this remains a subject of debate, studies of SNARE protein-mediated membrane fusion show that although hemifusion is often observed, it may function as a metastable trap [133,134].

Gui et al. [115] suggest that for influenza virus membrane fusion, activated HA first engages the target membrane through its exposed fusion peptide, forming bridging contacts between the two opposing membranes (Figure 4A). Subsequent HA refolding induces dimpling in the target membrane as it is drawn towards the matrix-reinforced viral membrane leading to formation of localized close contact zones between the two membranes, which the authors suggest may serve to minimize the initial energetic penalty incurred from dehydrating the membrane surface (Figure 4A,B) [40,123]. The contact zone then expands to form an extended, tightly docked interface between the two outer membrane leaflets (Figure 4A,B). Similar extended interface contacts have been observed during SNARE protein-mediated membrane fusion and the GTP-dependent alastin fusion protein [2,34,133,134,135,136,137]. Higher levels of cholesterol or inclusion of lipids found in mature endosomes in the target membrane were found to promote formation of these extended interfaces [115]. The tightly docked membranes transitioned to the post-fusion state thus supporting the authors’ conclusion that these extended interfaces are a critical stage along the fusion reaction [114,115,117].

Membrane composition contributes to how membrane fusion proceeds and what lipid organizations are populated and enriched during the process [40,113,114,117,120,121,123,138,139]. A separate recent cryo-ET study of influenza virus membrane fusion with synthetic liposomes, where the liposome composition differed from that used by Gui et al., found hemifused virus-liposome complexes in high abundance [114]. While both cryo-ET studies concluded that cholesterol is critically important for productive and complete fusion to occur, their conflicting results regarding the prevalence of hemifused membranes indicate that membrane composition can influence the pathway of membrane remodeling during fusion.

One aspect that makes clear the need to study whole virions to understand the fusion process relates to the role of the M1 matrix protein layer. Although classically HA has been considered the prime mediator of fusion, M1 likewise appears to play an important role in regulating the order of events as well as the ability for the virus membrane to deform and complete fusion [115]. At increasingly acidic conditions the previously well-ordered M1 matrix layer dissociates from the viral membrane conferring plasticity to the lipid bilayer and greater lateral mobility of the HAs engaged in fusion [113,140]. The HA transmembrane domain (TMD) and cytoplasmic tail are believed to interact with the M1 matrix layer, thus dissociation of the matrix layer and abolition of this interaction would free the HA TMD enabling it to mobilize and recombine with the HA2 fusion peptide as the fusion reaction proceeds to completion [113,141,142].

These cryo-ET studies present the most complete observations describing viral protein-mediated membrane fusion to date and, taken together with recent observations on the structural mechanics of HA-fusion activation, these studies show how new approaches are providing unprecedented views into mechanisms of protein-mediated membrane fusion.

## 7. Understanding HA-mediated Fusion as a First Step towards a Broader Understanding of Fusion in Diverse Enveloped Viruses

Influenza HA serves as an invaluable model system for studying the mechanism of fusion activation and protein-mediated membrane fusion. It is unclear, however, whether the mechanistic models derived from the study of the HA described above (a subtype H3 HA in the HDX-MS case and H5 in the sm-FRET example) are directly generalizable across all HA subtypes that exhibit different pH sensitivities and overall stability, or to other type I fusion proteins that are triggered by different signals [22,56]. During infection, the endosomal lumen becomes increasingly acidic as it matures [8,140,143,144]. Evidence suggests that the gradual and stepwise endosomal acidification is important for priming HA and the influenza virus for membrane fusion [81]. This dynamic structural priming is likely important for ensuring that influenza virus membrane fusion is triggered at the correct pH, and thus cellular location. The efficiency of membrane fusion by some HAs is highly dependent on pH, whereas other HAs, even within the same group, display seemingly no such dependence [144]. At present, we do not understand the structural basis for HA’s variable acid stability and how these differences manifest mechanistically during fusion activation.

Despite recent advances, our understanding of influenza virus membrane fusion is still incomplete. For example the route of viral uptake by cells is dependent on viral morphology [32]. Influenza virus is highly pleomorphic and ranges from small spherical virions ~100 nm in diameter to large filamentous virions up to 1 µm in length, and the balance of particle morphology to one end of the spectrum vs the other can vary dramatically with influenza strain [126,145]. At present, it is not fully understood how different entry pathways for large and small influenza particles influences the acidification pathway or the mechanics of HA-mediated membrane fusion, or how morphology influences engagement between the virus and target membranes [146,147]. It remains to be determined what the appropriate triggering conditions to mimic the in vitro trafficking of large vs small influenza particles are, and whether morphology itself may alter the influenza fusion pathway.

Beyond influenza, low pH activation of HA is relatively simple compared to other type I fusion proteins that are activated by receptor binding, proteolytic cleavage, environmental factors (such as low pH, temperature, or cations), and any combination of these factors [8]. Thus, the dynamic and staged activation of HA in response to increasingly low pH conditions may not be a generalizable phenomenon amongst type I fusion proteins. If we extend our consideration to Type II where in many cases the fusion proteins are organized symmetrically on the icosahedral particles, the ability of fusion proteins to work in concert may be quite different to the case of the influenza virus [148]. It will be fascinating to revisit the basic questions of: what is the nature of protein conformational changes that drive fusion, what happens to the membranes themselves, and how are protein and membrane remodeling coupled together for such systems when comparable biophysical and structural data are available in the future? What is clear at this stage, is that we now possess the tools that can address those fundamental questions of how proteins undergo conformational changes that mediate the process of membrane deformation and remodeling that ultimately lead to productive fusion. Thus, we are now poised to gain a far deeper understanding by directly probing, imaging, and structurally dissecting the mechanics and dynamic transitions that are at the heart of the process of protein-mediated membrane fusion.

## Figures and Tables

**Figure 1 viruses-12-00413-f001:**
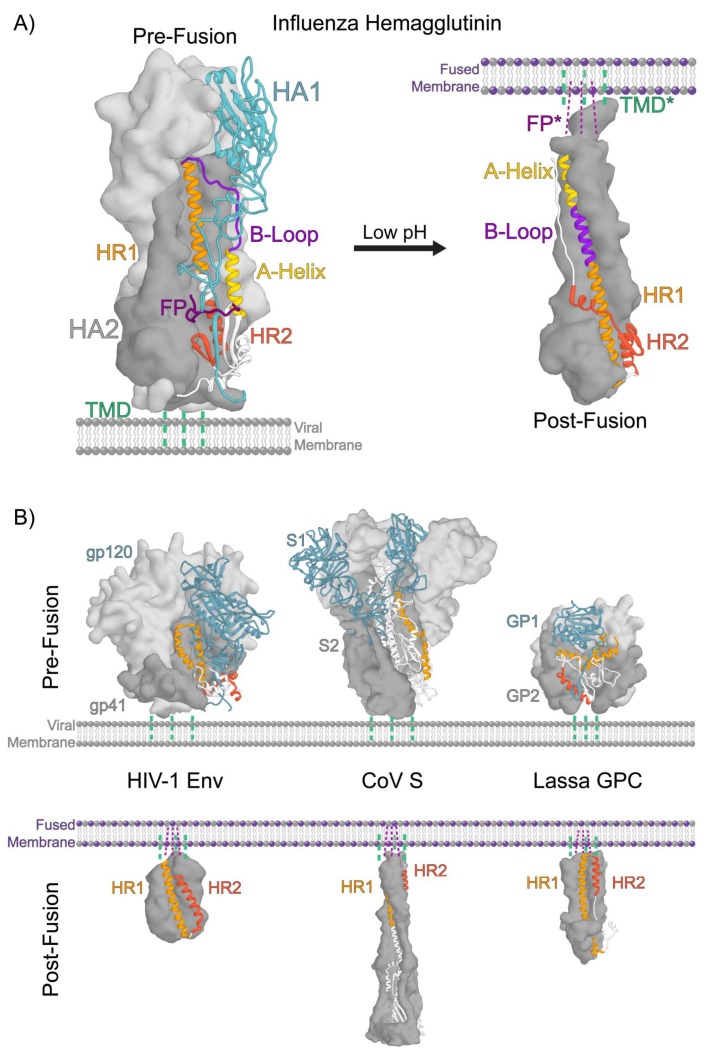
Architecture of a type I fusion protein. (**A**) The structures of the influenza hemagglutinin (HA) fusion protein in the pre-fusion (PDB 3HMG) and post-fusion (PDB 1QU1) states highlight the dramatic pH-dependent reorganization that drives the membrane fusion reaction. The pre-fusion state is metastable with respect to the post-fusion state. In the pre-fusion state, the HA1 receptor binding domain (RBD) (shown in light grey volume and blue ribbon) forms a “clamp” interaction with the HA2 fusion subunit, thereby stabilizing the high-energy, “spring-loaded” HA2 fusion domain (highlighted by the HA1-HA2 interface shown in grey). The HA2 N-terminal fusion peptide (FP shown in dark magenta) forms a “hook” within the fusion domain, lashing adjacent protomers together. Once destabilized by low pH, these interactions are lost and the HA reorganizes to the post-fusion state where the N-terminal FP and transmembrane domain (TMD) are colocalized in the newly fused membrane. The post-fusion state is characterized by the trimer of hairpins formed by the two heptad repeat regions (HR1 and HR2). (**B**) Comparison of the pre-fusion (top) and post-fusion (bottom) structures of diverse type I fusion proteins reveals the conservation of core architectural features including the “clamp” interaction between the RBD and fusion domain and reorganization of the two heptad repeats into a trimer of hairpins. Shown are the pre- and post-fusion structures of the HIV-1 Env (PDB 5FUU and 1I5X), Coronavirus (CoV) S (PDB 5W9J and 6B3O), and Lassa virus glycoprotein complex (GPC) (PDB 5VK2 and 5OMI).

**Figure 2 viruses-12-00413-f002:**
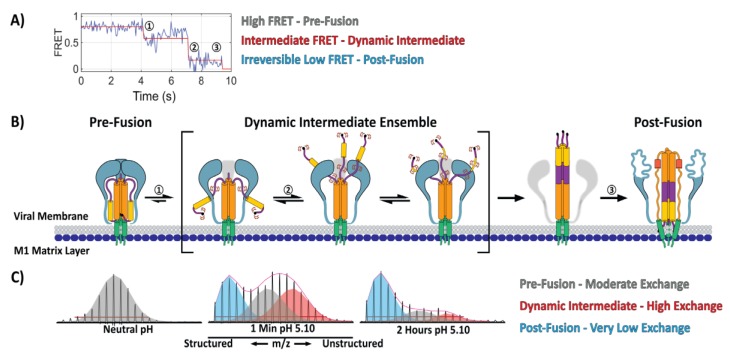
Structural mechanics of influenza-HA-fusion activation. (**A**) Single molecule FRET (sm-FRET) monitoring of HA during fusion activation shows the transition from the pre-fusion state (1) (high FRET) through an obligate and dynamic intermediate (2) (intermediate FRET and reversible low FRET) to the irreversible post-fusion state (3) (low FRET) (figure modified with permission from Das et al., 2018). (**B**) Cartoon model describing HA-fusion activation shows the formation of a dynamic intermediate ensemble, as supported by sm-FRET and hydrogen/deuterium-exchange mass spectrometry (HDX-MS). Transitions between states are labeled according to those observed by sm-FRET (panel A) and the dynamic intermediate state is depicted according to the HDX-MS study (panel C). (**C**) Pulse-labeling HDX-MS reveals the formation of a dynamic intermediate state in fusion-active HA on infectious influenza virions. In the neutral pH pre-fusion state (left—grey envelope) the HA2 B-loop peptide becomes labeled with a moderate level of deuterium as it is a structured loop in the pre-fusion state. After incubation at pH 5.10 for 1 min the B-loop displays three unique HDX states corresponding to the pre-fusion state (grey—moderate level of deuterium exchange), post-fusion helical bundle (blue—very low level of deuterium exchange), and dynamic intermediate (red—high level of deuterium exchange). After continued incubation at low pH the HA2 B-loop transitions monotonically to the post-fusion state (blue).

**Figure 3 viruses-12-00413-f003:**
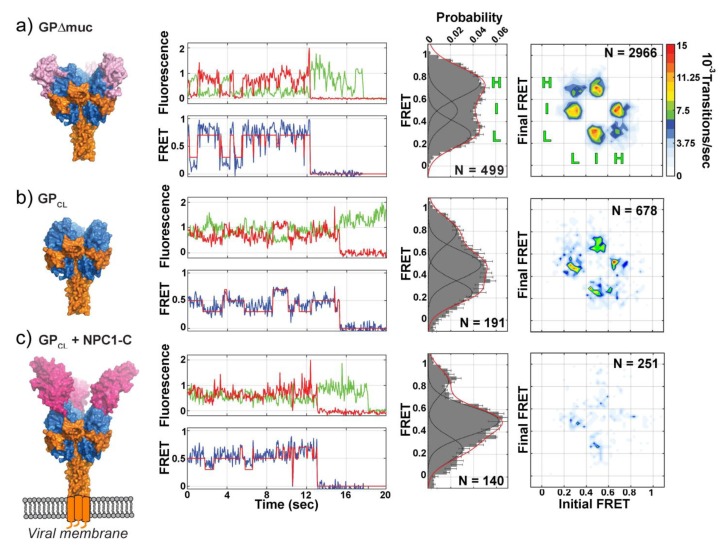
Dynamic conformational changes in Ebola virus GP during viral entry. (**A**) sm-FRET monitoring of Ebola GP without the mucin-like domain (GPΔmuc) revealed the GP was highly dynamic under equilibrium conditions and reversibly transitioned between three states: the high-FRET state (H) corresponding to the pre-fusion conformation, intermediate-FRET state (I), and low-FRET state (L). Population FRET histograms show the equilibrium distribution of all observed FRET states. Transition density plots (TDP) (far right) reveal the direct transitions between each FRET state for all observed trajectories. (**B**) Removal of the glycan cap resulted in lower occupancy of the high-FRET state and increased occupancy of the intermediate-FRET state. The glycan cap (GP_CL_) also displayed reduced transitions between all FRET states indicating lowered conformational and structural dynamics. (**C**) Niemann-Pick C1 (NPC1) receptor binding to GP_CL_ further biased the equilibrium distribution towards the intermediate-FRET state and reduced transitions between states. Receptor binding did not result in irreversible transitions but rather quenched the conformational dynamics of GP. Figure modified with permission from [105].

**Figure 4 viruses-12-00413-f004:**
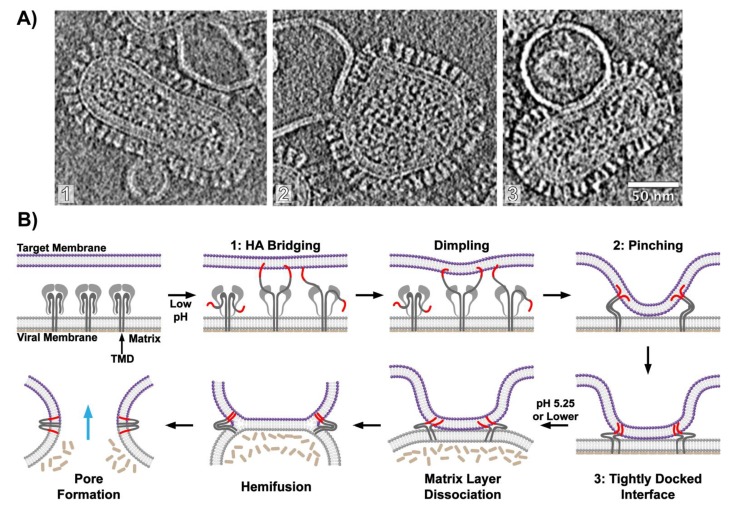
Visualizing influenza virus membrane fusion. Intermediates throughout the membrane fusion were visualized by cryo-electron tomography (cryo-ET) and the sequence of membrane remodeling was elucidated. (**A**) Cryo-ET images highlight key intermediates during membrane fusion including HA bridging (1), membrane pinching (2), and formation of a tightly docked interface (3) (left to right) (scale bar = 50 nm). (**B**) Cartoon model describing the membrane ultrastructure and sequence of intermediates during influenza viral membrane fusion.

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
