# Peer review of "New Biophysical Approaches Reveal the Dynamics and Mechanics of Type I Viral Fusion Machinery and Their Interplay with Membranes"

_viruses, 2020, doi:10.3390/v12040413_

Round 1
Reviewer 1 Report
Studies of viral fusion protein function have benefited greatly in the last decade by the development of new approaches, and the review by Benhaim and Lee presents the state of the field for influenza HA, a prototype for class I viral fusion protein, providing a nice resource for understanding our updated understanding of influenza HA fusion. The sections on influenza HA are well-written, but there is little on other systems, so it would seem appropriate to modify the title and abstract to reflect this.
Specific comments:
- On lines 108-110, the authors indicate the influenza HA is cleaved by extracellular proteases, but this is not true of all strains of HA. For accuracy, the existence of furin-cleaved HA molecules should also be mentioned.
- The review focuses almost exclusively on work on influenza HA, with the exception of a small section on comparing structures of other class I fusion proteins, and a section at the end on the recent work from the Munro group on ebola. Thus, it would seem most appropriate to modify the title and the abstract to indicate this is a review on HA fusion, not a broad review on all viral fusion proteins.
Author Response
First we would like to thank the reviewer for taking the time to carefully review our manuscript and for the responses. We have revised out statement on the proteolytic processing of HA0 to include those strains that are processed by furin (Line 106). After reviewing our manuscript we agree with the reviewer that it was light on information describing other type I fusion proteins and the current body of knowledge surrounding their activation mechanisms. We have made significant revisions to the manuscript in order to address these concerns and make the review more comprehensive and informative. Specifically, in Section 4 beginning on line 234 we have added sections discussing the HIV-1 Env fusion glycoprotein, Lassa virus GPC, and paramyxovirus f proteins and the fields current understanding of their mechanisms of activation and the difficulties associated with studying systems with more complex mechanisms than influenza HA. We have also expanded section 5 (beginning on line 353) to include a more expansive overview of the recent sm-FRET studies on the Ebola virus GP protein along with an associated figure for that section. We believe that these changes have resulted in a more inclusive and informative review that provides an overview of recent research discoveries and approaches.
Reviewer 2 Report
This is an important and much needed review of the most recent advances in the biophysical study of the changes that take place in the influenza hemagglutinin as it promotes membrane fusion, as well as the nature of the interaction of the protein with the membranes involved. The authors have produced a very comprehensive and timely summary of the state of our understanding of the intricacies of influenza-induced membrane fusion.
There are a few minor comments, which if addressed by the authors, might improve the manuscript:
- Since the manuscript focuses pretty much exclusively on HA-induced fusion, I feel the title should reflect this. Indeed, even the authors recognize that the knowledge reviewed here “may not be a generalizable phenomenon amongst class I fusion proteins”, indeed possibly not even across all HA subtypes (lines 441-2).
- The paramyxovirus tandem of the hemagglutinin-neuraminidase and fusion proteins are given short shrift here. Although the review does focus on influenza HA, the authors might want to mention these two proteins, in light of the fact that an interaction between the two mediates the triggering of fusion by receptor binding with these two functions contributed by different proteins.
- I feel that the sentence in lines 14-15 of the abstract are an overstatement. While I agree, as stated above, that these new biophysical approaches have significantly advanced the field, this sentence makes it seem like not much was known previously. There was already a substantial level of understanding about the conformational changes in fusion proteins and their effects on membranes based on prior crystallographic and structural studies.
Author Response
First we would like to thank the reviewer for taking the time to carefully review our manuscript and for their responses. After reviewing our manuscript we agree with the reviewer that the review did not sufficiently discuss viral fusion proteins beyond HA and their mechanisms of fusion activation. To this end we have made significant revisions to the manuscript that we believe strengthen the main points of the review and provide a more comprehensive overview of our current understanding of all type I fusion proteins. Specifically, in Section 4 beginning on line 234 we have added sections discussing the HIV-1 Env fusion glycoprotein, Lassa Virus GPC, and paramyxovirus f proteins and the fields current understanding of their mechanisms of activation and the difficulties associated with studying systems with more complex mechanisms than influenza HA. Furthermore, we have greatly expanded section 5 (beginning on line 353) to include a more expansive overview of the recent sm-FRET studies on the Ebola virus GP protein along with an associated figure for that section. We have also revised our statement addressing the generalizability of influenza HAs activation mechanism to include a comparison to all other HAs as well (Line 521).
We appreciate and understand the reviewer’s position on the statement in our abstract (lines 14-15) pertaining to the prior characterization of the conformational changes that occur during fusion activation and membrane fusion and their effect on the membranes. We have updated this wording to more specifically reference structural and dynamic aspects of changes in fusion proteins.